# Boolean Decision Rules via Column Generation

**Sanjeeb Dash, Oktay Günlük, Dennis Wei**
IBM Research
Yorktown Heights, NY 10598, USA
{sanjeebd,gunluk,dwei}@us.ibm.com

## Abstract

This paper considers the learning of Boolean rules in either disjunctive normal form (DNF, OR-of-ANDs, equivalent to decision rule sets) or conjunctive normal form (CNF, AND-of-ORs) as an interpretable model for classification. An integer program is formulated to optimally trade classification accuracy for rule simplicity. Column generation (CG) is used to efficiently search over an exponential number of candidate clauses (conjunctions or disjunctions) without the need for heuristic rule mining. This approach also bounds the gap between the selected rule set and the best possible rule set on the training data. To handle large datasets, we propose an approximate CG algorithm using randomization. Compared to three recently proposed alternatives, the CG algorithm dominates the accuracy-simplicity trade-off in 8 out of 16 datasets. When maximized for accuracy, CG is competitive with rule learners designed for this purpose, sometimes finding significantly simpler solutions that are no less accurate.

## 1 Introduction

Interpretability has become a well-recognized goal for machine learning models. The need for interpretable models is certain to increase as machine learning pushes further into domains such as medicine, criminal justice, and business, where such models complement human decision-makers and decisions can have major consequences on human lives. Transparency is thus required for domain experts to understand, critique, and trust models, and reasoning is required to explain individual decisions.

This paper considers Boolean rules in either disjunctive normal form (DNF, OR-of-ANDs) or conjunctive normal form (CNF, AND-of-ORs) as a class of interpretable models for binary classification. An example of a DNF rule with two clauses is "IF (# accounts $< 5$) OR (# accounts $\geq 7$ AND debt $> \$1000$) THEN risk = high". Particularly desirable for interpretability are compact Boolean rules with few clauses and conditions in each clause.

DNF classification rules are also referred to as decision rule sets, where each conjunction is considered an individual rule, rules are unordered, and a positive prediction is made when at least one of the rules is satisfied. Rule sets stand in contrast to decision lists [44, 35, 49, 3, 34, 53], where rules are ordered in an IF-ELSE sequence, and decision trees [11, 43, 6], where they are organized into a tree structure. While the latter two classes are also considered interpretable, the metrics for measuring their complexity are different and not directly comparable [27]. Moreover, a user study [33] has quantified the extra effort involved in understanding decision lists due to the need to account for the negations of all preceding rules.

The learning of Boolean rules and rule sets has an extensive history spanning multiple fields. DNF learning theory (e.g. [47, 32, 24]) focuses on the ideal noiseless setting (sometimes allowing arbitrary queries) and is less relevant to the practice of learning compact models from noisy data. Predominant practical approaches include a covering or separate-and-conquer strategy ([15, 14, 16, 26, 28, 40],

see also the survey [30]) of learning rules one by one and removing "covered" examples, a bottom-up strategy of combining more specific rules into more general ones [45, 22, 41], and associative classification in which association rule mining is followed by rule selection using various criteria [38, 36, 54, 50, 12, 13]. Broadly speaking, these approaches employ heuristics and/or multiple criteria not directly related to classification accuracy. Moreover, they do not explicitly consider model complexity, a problem that has been noted especially with associative classification. Rule set models have been generalized to rule ensembles [17, 29, 20], using boosting and linear combination rather than logical disjunction; the interpretability of such models is again not comparable to rule sets. Models produced by logical analysis of data [9, 31] from the operations research community are similarly weighted linear combinations.

In recent years, spurred by the demand for interpretable models, several researchers have revisited Boolean and rule set models and proposed methods that jointly optimize accuracy and simplicity within a single objective function. These works however have both restricted the problem and approximated its solution. In [33, 52, 51], frequent rule miners are first used to produce a set of candidate rules. A greedy forward-backward algorithm [33], simulated annealing [52], or integer programming (IP) (in an unpublished manuscript [51]) are then used to select rules from the candidates. The drawback of rule mining is that it limits the search space while often still producing a large number of rules, which then have to be filtered using criteria such as information gain. [51] also presented an IP formulation (but no computational results) that jointly constructs and selects rules without pre-mining. [46] developed an IP formulation for DNF and CNF learning in which the number of clauses (conjunctions or disjunctions) is fixed. The problem is then solved approximately by decomposing into subproblems and applying a linear programming (LP) method [39], which requires rounding of fractional solutions.

In this paper, we also propose an IP formulation for Boolean rule (DNF or CNF) learning but one that avoids the above limitations. Rather than mining rules, we use the large-scale optimization technique of column generation (CG) to intelligently search over the exponential number of all possible clauses, without enumerating even a pre-mined subset (which can be large). Instead, only those clauses that can improve the current solution are generated on the fly. In practice, our approach solves the IP formulation to provable optimality for smaller datasets. For large datasets we employ an approximate version of CG by randomly selecting samples and candidate features that can be used in a clause. To speed up computation, we also generate additional clauses using a greedy algorithm that still optimizes the correct objective.

A numerical evaluation is presented using 16 datasets, including one from the ongoing FICO Explainable Machine Learning Challenge [1]. In terms of the trade-off achieved between accuracy and rule simplicity, our CG algorithm dominates three other recent proposals on 8 datasets, whereas each of the others dominates on at most two. When optimized for accuracy using cross-validation, CG remains competitive with rule learners such as RIPPER [16] that are designed for maximum accuracy. In some instances it provides significantly less complex models with no sacrifice in accuracy.

We note that CG has been proposed for other machine learning tasks such as boosting [21, 7] and hash learning [37]. In [21] however, the pricing problem (see Section 2.2) is solved approximately by a weak learning algorithm ("weak" in the boosting sense), not IP, whereas in [7], pricing can be done tractably through enumeration.

## 2 Problem formulation

We consider supervised binary classification given a training dataset of $n$ samples $(\mathbf{x}_i, y_i)$, $i = 1, \ldots, n$ with labels $y_i \in \{0, 1\}$. Let the set $\{1, \ldots, n\}$ be partitioned into $\mathcal{P} \cup \mathcal{Z}$ where $\mathcal{P}$ contains the indices of the samples with label $y_i = 1$ and $\mathcal{Z}$ contains the ones with label $y_i = 0$. For the problem formulation in this section, all features $X_j$, $j \in \mathcal{J} = \{1, \ldots, d\}$, are assumed to be binary-valued as well; binarization of numerical and categorical features is discussed in Section 4.

The presentation focuses on the problem of learning a Boolean classifier $\hat{y}(\mathbf{x})$ in DNF (OR-of-ANDs). Given a DNF and binary-valued features, a clause corresponds to a conjunction of features and a sample satisfies a clause if it has all features contained in the clause (i.e. $x_{ij} = 1$ for all such features $j$). Since a DNF classifier is equivalent to a rule set, the terms clause, conjunction, and (single) rule (within a rule set) are used interchangeably. As shown in [46] using De Morgan's laws, the same

formulation applies equally well to CNF learning by negating both labels $y_i$ and features $\mathbf{x}_i$. The method can also be extended to multi-class classification in the usual one-versus-rest manner.

## 2.1 An integer program to minimize Hamming loss

Our objective is to minimize the *Hamming loss* of the rule set as is also done in [46, 33]. For each incorrectly classified sample, the Hamming loss counts the number of clauses that have to be selected or removed to classify it correctly. More precisely, it is equal to the number of samples with label 1 that are classified incorrectly (false negatives) plus the sum of the number of selected clauses that each sample with label 0 satisfies. Thus while each false negative contributes one unit to this loss function, representing a single clause that needs to be selected, a false positive would contribute more than one unit if it satisfies multiple clauses, which must all be removed.

We bound the *complexity* of the rule set by a given parameter $C$, both to prevent over-fitting and to control complexity. For concreteness, we define the complexity of a clause to be a fixed cost of one plus the number of conditions in the clause; other linear combinations can be handled equally well. The total complexity of a rule set is defined as the sum of the complexities of its clauses. Alternatively, it is possible to include an additional term in the objective function to penalize complexity but we find it more natural to explicitly bound the maximum complexity as it can offer better control in applications where interpretable rules are preferred. Clearly it is also possible to use both a constraint and a penalty term.

We express the above notions of Hamming loss and complexity in an integer program (IP) that is not practical for real-life datasets as written but is useful to explain the conceptual framework behind our approach. Let $\mathcal{K}$ denote the collection of all possible (exponentially many) clauses involving $X_j$, $j \in \mathcal{J}$ and $\mathcal{K}_i \subseteq \mathcal{K}$ contain the clauses satisfied by sample $i$ for all $i \in \mathcal{P} \cup \mathcal{Z}$. Note that as the features $X_j$ are binary, $|\mathcal{K}|$ is indeed bounded. Letting decision variable $w_k$ for $k \in \mathcal{K}$ denote whether clause $k$ is used in the rule set, $c_k$ denote the complexity of clause $k \in \mathcal{K}$, and $\xi_i$ for $i \in \mathcal{P}$ denote the positive samples classified incorrectly, we have the following IP:

$$z_{MIP} = \mathbf{min} \quad \sum_{i \in \mathcal{P}} \xi_i + \sum_{i \in \mathcal{Z}} \sum_{k \in \mathcal{K}_i} w_k \tag{1}$$

$$\mathbf{s.t.} \quad \xi_i + \sum_{k \in \mathcal{K}_i} w_k \geq 1, \quad \xi_i \geq 0, \quad i \in \mathcal{P} \tag{2}$$

$$\sum_{k \in \mathcal{K}} c_k w_k \leq C \tag{3}$$

$$w_k \in \{0, 1\}, \quad k \in \mathcal{K}. \tag{4}$$

The objective function (1) is the Hamming loss as described. Constraints (2) identify false negatives, which have $\sum_{k \in \mathcal{K}_i} w_k = 0$ and are therefore not "covered" by any selected clauses. Note that $w_k$ being binary implies that $\xi_i \in \{0, 1\}$ in any optimal solution because of the objective function. Constraint (3) bounds the complexity of the rule set. We call this formulation the *Master IP* (MIP) and call its linear programming (LP) relaxation, obtained by dropping the integrality constraint (4), the *Master LP* (MLP), denoting its optimal value by $z_{MLP}$. It is also possible to weight the two terms in the objective (1) differently, for example to balance unequal classes, but we do not pursue that variation here.

## 2.2 Column generation framework

Clearly it is only practical to solve the Master IP for very small datasets. Moreover, even solving the Master LP explicitly is often intractable due to the fact that it has exponentially many variables. An effective way to solve such large LPs is to use the *column generation* framework [4, 18] where only a small subset of all possible $w_k$ variables (clauses) is generated explicitly and the optimality of the LP is guaranteed by iteratively solving a *pricing problem*.

To apply this framework to the MIP, the first step is to restrict the formulation by replacing the set $\mathcal{K}$ with a very small subset of it and explicitly solve the LP relaxation of the resulting smaller problem, which we call the *Restricted MLP*. Any optimal solution of the Restricted MLP can be extended to a solution of MLP with the same objective value by setting all missing $w_k$ variables to zero, and thus provides an upper bound on $z_{MLP}$. Such a solution can potentially be improved by augmenting the

Restricted MLP with additional variables corresponding to some of the missing clauses. The second step is to identify such clauses without explicitly considering all of them. Repeating these steps until there are no improving clauses (i.e. variables missing from the Restricted MLP that can reduce the cost) solves the MLP to optimality.

To find the missing clauses that can potentially improve the value of the Restricted MLP, one needs to check if there are variables missing from the Restricted MLP that have negative reduced cost [5]. The reduced cost of a missing variable gives the maximum possible change in objective value per unit increase in that variable's value (when it is included in the formulation). Therefore, if all missing variables have non-negative reduced cost, then the current Restricted MLP cannot be improved and its optimal solution yields an optimal solution of the MLP. Furthermore, it is desirable to identify missing variables that have large negative reduced costs as they are more likely to improve the objective value of the Restricted MLP. To this end, we next formulate an optimization problem that uses the optimal dual solution to the Restricted MLP. Let $\mu_i \geq 0$ for $i \in \mathcal{P}$ denote the dual variables associated with constraints (2) and $\lambda \geq 0$ be the dual variable associated with (3). Let $\delta_i \in \{0, 1\}$ denote whether the $i$th sample satisfies a missing clause in question. If we let $c$ denote the complexity of the clause, then its reduced cost is equal to

$$\sum_{i \in \mathcal{Z}} \delta_i - \sum_{i \in \mathcal{P}} \mu_i \delta_i + \lambda c. \tag{5}$$

The first term in (5) is the cost of the missing clause in the objective function (1), expressed in terms of $\delta_i$. The second term is the sum of the dual variables associated with constraints (2) in which the clause appears. The last term is the dual variable associated with constraint (3) multiplied by the complexity of the clause.

We now formulate an IP to express clauses as conjunctions of the original features $X_j$, $j \in \mathcal{J}$. Let the decision variable $z_j \in \{0, 1\}$ denote if feature $j \in \mathcal{J}$ is selected in the clause. Let $S_i$ correspond to the zero-valued features in sample $i \in \mathcal{P} \cup \mathcal{Z}$, $S_i = \{j : x_{ij} = 0\}$. Then the *Pricing Problem* below identifies the clause missing from the Restricted MLP that has the lowest reduced cost.

$$z_{CG} = \min \quad \lambda \left( 1 + \sum_{j \in J} z_j \right) - \sum_{i \in \mathcal{P}} \mu_i \delta_i + \sum_{i \in \mathcal{Z}} \delta_i \tag{6}$$

$$\text{s.t.} \qquad \delta_i + z_j \leq 1, \qquad\qquad j \in S_i, \ i \in \mathcal{P} \tag{7}$$

$$\delta_i \geq 1 - \sum_{j \in S_i} z_j, \quad \delta_i \geq 0, \qquad i \in \mathcal{Z} \tag{8}$$

$$\sum_{j \in J} z_j \leq D, \tag{9}$$

$$z_j \in \{0, 1\}, \qquad\qquad j \in J. \tag{10}$$

The first term in (6) expresses the complexity $c_k$ in terms of the number of selected features. Constraints (7), (8) ensure that the clause acts as a conjunction, i.e. it is satisfied ($\delta_i = 1$) only if no zero-valued features are selected ($z_j = 0$ for $j \in S_i$). Similar to $\xi_i$ in MIP, the variables $\delta_i$ do not have to be explicitly defined as binary due to the objective function. Constraint (9) bounds the number of features allowed in any clause in the rule set. Parameter $D$ above can be set to $C - 1$ to relax this constraint, or it can be set to a smaller number if desired to limit the clause complexity.

The optimal solution to the Pricing Problem above gives the clause with the minimum reduced cost that is missing from the Restricted MLP. The reduced cost of this clause equals $z_{CG}$ and if $z_{CG} < 0$, then the corresponding variable is added to the Restricted MLP. More generally, any feasible solution to the Pricing Problem that has a negative objective function value gives a clause with a negative reduced cost and therefore can be added to the Restricted Restricted MLP to improve its value.

## 2.3 Optimality guarantees and bounds

When the column generation framework described above is repeated until $z_{CG} \geq 0$, none of the variables missing from the Restricted MLP have a negative reduced cost and the optimal solution of the MLP and the Restricted MLP coincide. In addition, if the optimal solution of the Restricted MLP turns out to be integral, then it is also an optimal solution to the MIP and therefore MIP is solved

to optimality. If the optimal solution of the Restricted MLP is fractional, then one may have to use column generation within an enumeration framework to solve MIP to optimality. This approach is called *branch-and-price* [4] and is quite computationally intensive.

However, even when the optimal solution to the MLP is fractional, $\lceil z_{MLP} \rceil$ provides a lower bound on $z_{MIP}$ as the objective function (1) has integer coefficients. This lower bound can be compared to the cost of any feasible solution to MIP. If the latter equals $\lceil z_{MLP} \rceil$, then, once again, MIP is solved to optimality. As one example, a feasible solution to MIP could be obtained by solving the *Restricted MIP* obtained by imposing (4) on the variables present in the Restricted MLP. More generally, any heuristic method can generate feasible solutions to MIP.

Finally, we note that even when the MLP is not solved to optimality and the column generation procedure is terminated prematurely, a valid lower bound on $z_{MIP}$ can be obtained by $\lceil z_{RMLP} + (C/2)z_{CG} \rceil$, where $z_{RMLP}$ is the objective value of the last Restricted MLP solved to optimality. This bound is due to the fact that $c_k \geq 2$ for any clause and there might be at most $C/2$ missing variables with reduced cost no less than $z_{CG}$ that can be added to the Restricted MLP [48].

## 3 Computational Approach

The previous section provides a sound theoretical framework for finding an optimal rule set for the training data. For *small* datasets, defined loosely as having less than a couple of thousand samples and less than a few hundred binary (binarized) features (this includes the mushroom and tic-tac-toe UCI datasets appearing in Section 4), it is computationally feasible to employ this optimization framework as described in Section 2. However, to handle larger datasets within a time limit of 10 or 20 minutes, one has to sacrifice the optimality guarantees of the framework. We next describe our computational approach to deal with larger datasets, which can be seen as an optimization-based heuristic. We call a dataset *medium* if it has more than a couple of thousand samples but less than a few hundred binary features. We call it *large* if it has many thousands of samples and more than several hundred binary features. The separation of datasets into small, medium and large is done based on empirical experiments to improve the likelihood that the Pricing Problem can produce negative reduced cost solutions.

For medium and large datasets, the number of non-zeros in the Pricing Problem (defined as the sum of the numbers of variables appearing in the constraints of the formulation) is at least 100,000 and solving this integer problem in a reasonable amount of time is not always feasible. Consequently solving the MLP to proven optimality is not likely. To deal with this practical issue, we terminate the Pricing problem if a fixed time limit is exceeded. We use a standard mixed-integer programming solver (CPLEX 12.7.1) to which a time limit can be provided.

While the solver is finding negative reduced cost clauses from the Pricing Problem, the presence of the time limit matters little. If the Pricing Problem is solved to optimality within the time limit, then we obtain a minimum reduced cost clause. Moreover, the solver might discover several negative reduced cost clauses within the time limit and it is possible to recover all these solutions at termination (due to optimality or time limit). To speed up the overall solution process, we add all the negative reduced cost clauses returned by the solver to the Restricted MLP. As long as one variable with a negative reduced cost is obtained, the column generation process continues.

Eventually, the solver will fail to find a negative reduced cost solution within the time limit. If the solver proves that there is no such solution to the Pricing Problem, then the MLP is solved to optimality. However, if non-existence cannot be proved within the time limit, then column generation using the Pricing Problem has to terminate without an optimality guarantee or a valid lower bound on the MIP. In this case, we employ a fast heuristic algorithm to continue to search for negative reduced cost solutions and extend the process.

Our heuristic algorithm only explores clauses that have up to $\kappa$ features (we use $\kappa = 5$ in our experiments), and is as follows. We create all one-term clauses that can be potentially extended to negative reduced cost clauses, and then assign each of them a score that equals the objective function of the Pricing problem applied to the clause. For each clause size $l$ from 1 to $\kappa$, we do the following: we process all generated clauses that have $l$ features in increasing order of their score, and for each such clause we create new clauses by appending additional features. Whenever we find a clause with negative reduced cost, we add it to a potential list of solutions, and then when our enumeration

terminates (we have an upper bound on the number of generated clauses), we return the best clauses generated by the heuristic before proceeding to the next value of $l$.

In addition to the time limit on the Pricing Problem, we also have a time limit on the overall column generation process. Thus column generation terminates in two cases: 1) when an improving clause cannot be found, either because one is proven not to exist or because one cannot be found within the Pricing Problem time budget and the heuristic also fails to find one, or 2) when the overall time limit is met. At this point, we solve the Restricted MIP (the integral version of the Restricted MLP) using CPLEX, and use the solution as our classifier.

For large datasets, the Pricing Problem can have more than a million non-zeros and even solving its LP relaxation becomes challenging. In this case the solver can rarely produce any negative reduced-cost solutions within the time limit. To deal with this, we formulate an approximate Pricing Problem by randomly selecting a limited number of features and samples. We pick samples uniformly with a probability that on average leads to a formulation with a couple of thousand samples. If the resulting Pricing Problem has more than a hundred thousand non-zeros, then we also limit the candidate features that can form a clause. The candidate features are selected uniformly with a probability that leads to a formulation with one hundred thousand non-zeros. We also note that for large datasets the Restricted MLP can easily have more than one million non-zeros after generating several hundred columns and it is faster to solve it with the interior point algorithm in CPLEX instead of simplex .

## 4 Numerical Evaluation

Evaluations were conducted on 15 classification datasets from the UCI repository [23] that have been used in recent works on rule set/Boolean classifiers [39, 19, 46, 52]. In addition, we used recently released data from the FICO Explainable Machine Learning Challenge [1]. It contains 23 numerical features of the credit history of $10,459$ individuals (9871 after removing records with all entries missing) for predicting repayment risk (good/bad). The domain of financial services and the clear meanings of the features combine to make it a good candidate for a rule set model. Details of how missing and special values were treated can be found in the supplementary material (SM). Test performance on all datasets is estimated using 10-fold stratified cross-validation (CV).

For comparison with our column generation (CG) algorithm, we considered three recently proposed alternatives that also aim to control rule complexity: Bayesian Rule Sets (BRS) [52] and the alternating minimization (AM) and block coordinate descent (BCD) algorithms from [46]. Additional comparisons include the WEKA [25] JRip implementation of RIPPER [16], a rule set learner that is still state-of-the-art in accuracy, and scikit-learn [42] implementations of the decision tree learner CART [11] and Random Forests (RF) [10]. The last is an uninterpretable model intended as a benchmark for accuracy. The SM includes further comparisons to logistic regression (LR) and support vector machines (SVM). The parameters of BRS and FPGrowth [8], the frequent rule miner that BRS relies on, were set as recommended in [52] and the associated code (see SM for details). For AM and BCD, the number of clauses was fixed at 10 with the option to disable unused clauses; initialization and BCD updating are done as in [46]. While both [46] and our method are equally capable of learning CNF rules, for these experiments we restricted both to learning DNF rules only.

We also experimented with code made available by the authors of [33]. Unfortunately, we were unable to execute this code with practical running time when the number of mined candidate rules exceeded 1000. Furthermore, the code was primarily designed to handle the interval representation of numerical features and not $(\leq, >)$ comparisons (see next paragraph). These limitations prevented us from making a full comparison. The SM includes partial results from [33] that are inferior to those from the other methods.

We used standard "dummy"/"one-hot" coding to binarize categorical variables into multiple $X_j = x$ indicators, one for each category $x$, as well as their negations $X_j \neq x$. For numerical features, there are two common approaches. The first is to discretize by binning into intervals and then encode as above with categorical features. The second is to compare with a sequence of thresholds, again including negations (e.g. $X_j \leq 1$, $X_j \leq 2$ and $X_j > 1$, $X_j > 2$). For these experiments, we used the second comparison method, as also recommended in [52, 46], with sample deciles as thresholds. Furthermore, features were binarized in the same way for all classifiers in this comparison, which all rely on discretization (but not for LR and SVM in the SM). Thus the evaluation controls for binarization method in addition to using the same training-test splits for all classifiers.

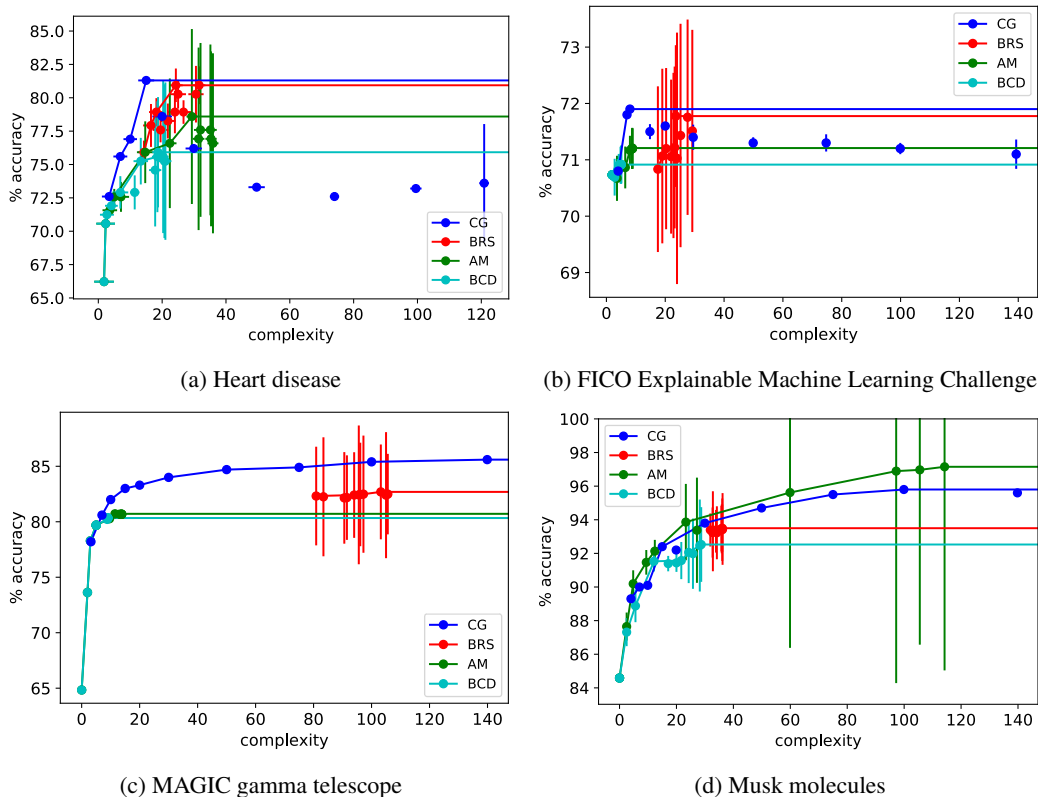

(a) Heart disease

(b) FICO Explainable Machine Learning Challenge

(c) MAGIC gamma telescope

(d) Musk molecules

Figure 1: Rule complexity-test accuracy trade-offs on 4 datasets. Pareto efficient points are connected by line segments. Horizontal and vertical bars represent standard errors in the means. Overall, the proposed CG algorithm dominates the others on 8 of 16 datasets (see the SM for the full set).

We first evaluated the accuracy-simplicity trade-offs achieved by our CG algorithm as well as BRS, AM, and BCD, methods that explicitly perform this trade-off. For CG, we used an overall time limit of 300 seconds for training and a time limit of 45 seconds for solving the Pricing Problem in each iteration. Low time limits were chosen partly due to practical considerations of running the algorithm multiple times (e.g. for CV) on many datasets, and partly to demonstrate the viability of IP with limited computation. As in Section 2, complexity is measured as the number of rules in the rule set plus the total number of conditions in the rules. For each algorithm, the parameter controlling model complexity (bound $C$ in (3), regularization parameter $\theta$ in [46], multiplier $\kappa$ in prior hyperparameter $\beta_l = \kappa|\mathcal{A}_l|$ from [52]) is varied, resulting in a set of complexity-test accuracy pairs. A sample of these plots is shown in Figure 1 with the full set in the SM. Line segments connect points that are Pareto efficient, i.e., not dominated by solutions that are more accurate and at least as simple or vice versa. CG dominates the other algorithms in 8 out of 16 datasets in the sense that its Pareto front is consistently higher; it nearly does so on a 9th dataset (tic-tac-toe) and on a 10th (banknote), all algorithms are very similar. BRS, AM, and BCD each achieve (co-)dominance only one or two times, e.g. in Figure 1d for AM. Among cases where CG does not dominate are the highest-dimensional datasets (musk and gas, although for the latter CG does attain the highest accuracy given sufficient complexity) and ones where AM and/or BCD are more accurate at the lowest complexities. BRS solutions tend to cluster in a narrow range despite varying $\kappa$ from $10^{-3}$ to $10^3$.

In a second experiment, nested CV was used to select values of $C$ for CG and $\theta$ for AM, BCD to maximize accuracy on each training set. The selected model was then applied to the test set. In these experiments, CG was given an overall time limit of 120 seconds for each candidate value of $C$ and the time limit for the Pricing Problem was set to 30 seconds. To offset for the decrease in the time limit, we performed a second pass for each dataset solving the restricted MIP with all the clauses generated for all possible choices of $C$. Mean test accuracy (over 10 partitions) and rule set complexity are reported in Tables 1 and 2. For BRS, we fixed $\kappa = 1$ as optimizing $\kappa$ did not improve

Table 1: Mean test accuracy (%, standard error in parentheses). **Bold**: Best among interpretable models; *Italics*: Best overall.

| dataset | CG | BRS | AM | BCD | RIPPER | CART | RF |
|---|---|---|---|---|---|---|---|
| banknote | 99.1 (0.3) | 99.1 (0.2) | 98.5 (0.4) | 98.7 (0.2) | **99.2** (0.2) | 96.8 (0.4) | *99.5* (0.1) |
| heart | 78.9 (2.4) | 78.9 (2.4) | 72.9 (1.8) | 74.2 (1.9) | 79.3 (2.2) | **81.6** (2.4) | *82.5* (0.7) |
| ILPD | 69.6 (1.2) | 69.8 (0.8) | **71.5** (0.1) | **71.5** (0.1) | 69.8 (1.4) | 67.4 (1.6) | 69.8 (0.5) |
| ionosphere | 90.0 (1.8) | 86.9 (1.7) | 90.9 (1.7) | **91.5** (0.7) | 88.0 (1.9) | 87.2 (1.8) | *93.6* (0.7) |
| liver | **59.7** (2.4) | 53.6 (2.1) | 55.7 (1.3) | 51.9 (1.9) | 57.1 (2.8) | 55.9 (1.4) | *60.0* (0.8) |
| pima | 74.1 (1.9) | **74.3** (1.2) | 73.2 (1.7) | 73.4 (1.7) | 73.4 (2.0) | 72.1 (1.3) | *76.1* (0.8) |
| tic-tac-toe | *100.0* (0.0) | 99.9 (0.1) | 84.3 (2.4) | 81.5 (1.8) | 98.2 (0.4) | 90.1 (0.9) | 98.8 (0.1) |
| transfusion | 77.9 (1.4) | 76.6 (0.2) | 76.2 (0.1) | 76.2 (0.1) | **78.9** (1.1) | 78.7 (1.1) | 77.3 (0.3) |
| WDBC | 94.0 (1.2) | 94.7 (0.6) | **95.8** (0.5) | **95.8** (0.5) | 93.0 (0.9) | 93.3 (0.9) | *97.2* (0.2) |
| adult | 83.5 (0.3) | 81.7 (0.5) | 83.0 (0.2) | 82.4 (0.2) | **83.6** (0.3) | 83.1 (0.3) | *84.7* (0.1) |
| bank-mkt | *90.0* (0.1) | 87.4 (0.2) | *90.0* (0.1) | 89.7 (0.1) | 89.9 (0.1) | 89.1 (0.2) | 88.7 (0.0) |
| gas | 98.0 (0.1) | 92.2 (0.3) | 97.6 (0.2) | 97.0 (0.3) | **99.0** (0.1) | 95.4 (0.1) | *99.7* (0.0) |
| magic | **85.3** (0.3) | 82.5 (0.4) | 80.7 (0.2) | 80.3 (0.3) | 84.5 (0.3) | 82.8 (0.2) | *86.6* (0.1) |
| mushroom | *100.0* (0.0) | 99.7 (0.1) | 99.9 (0.0) | 99.9 (0.0) | *100.0* (0.0) | 96.2 (0.3) | 99.9 (0.0) |
| musk | 95.6 (0.2) | 93.3 (0.2) | **96.9** (0.7) | 92.1 (0.2) | 95.9 (0.2) | 90.1 (0.3) | 86.2 (0.4) |
| FICO | 71.7 (0.5) | 71.2 (0.3) | 71.2 (0.4) | 70.9 (0.4) | **71.8** (0.2) | 70.9 (0.3) | *73.1* (0.1) |

accuracy on the whole (as can be expected from Figure 1). Tables 1 and 2 also include results from RIPPER, CART, and RF. We tuned the minimum number of samples per leaf for CART and RF, used 100 trees for RF, and otherwise kept the default settings. The complexity values for CART result from a straightforward conversion of leaves to rules (for the simpler of the two classes) and are meant only for rough comparison.

Table 2: Mean complexity (# clauses + total # conditions, standard error in parentheses)

| dataset | CG | BRS | AM | BCD | RIPPER | CART |
|---|---|---|---|---|---|---|
| banknote | 25.0 (1.9) | 30.4 (1.1) | 24.2 (1.5) | **21.3** (1.9) | 28.6 (1.1) | 51.8 (1.4) |
| heart | **11.3** (1.8) | 24.0 (1.6) | 11.5 (3.0) | 15.4 (2.9) | 16.0 (1.5) | 32.0 (8.1) |
| ILPD | 10.9 (2.7) | 4.4 (0.4) | **0.0** (0.0) | **0.0** (0.0) | 9.5 (2.5) | 56.5 (10.9) |
| ionosphere | 12.3 (3.0) | **12.0** (1.6) | 16.0 (1.5) | 14.6 (1.4) | 14.6 (1.2) | 46.1 (4.2) |
| liver | 5.2 (1.2) | 15.1 (1.3) | 8.7 (1.8) | **4.0** (1.1) | 5.4 (1.3) | 60.2 (15.6) |
| pima | 4.5 (1.3) | 17.4 (0.8) | 2.7 (0.6) | **2.1** (0.1) | 17.0 (2.9) | 34.7 (5.8) |
| tic-tac-toe | 32.0 (0.0) | 32.0 (0.0) | 24.9 (3.1) | **12.6** (1.1) | 32.9 (0.7) | 67.2 (5.0) |
| transfusion | 5.6 (1.2) | 6.0 (0.7) | **0.0** (0.0) | **0.0** (0.0) | 6.8 (0.6) | 14.3 (2.3) |
| WDBC | 13.9 (2.4) | 16.0 (0.7) | **11.6** (2.2) | 17.3 (2.5) | 16.8 (1.5) | 15.6 (2.2) |
| adult | 88.0 (11.4) | 39.1 (1.3) | 15.0 (0.0) | **13.2** (0.2) | 133.3 (6.3) | 95.9 (4.3) |
| bank-mkt | 9.9 (0.1) | 13.2 (0.6) | 6.8 (0.7) | **2.1** (0.1) | 56.4 (12.8) | 3.0 (0.0) |
| gas | 123.9 (6.5) | **22.4** (2.0) | 62.4 (1.9) | 27.8 (2.5) | 145.3 (4.2) | 104.7 (1.0) |
| magic | 93.0 (10.7) | 97.2 (5.3) | 11.5 (0.2) | **9.0** (0.0) | 177.3 (8.9) | 125.5 (3.2) |
| mushroom | 17.8 (0.3) | 17.5 (0.4) | 15.4 (0.6) | 14.6 (0.6) | 17.0 (0.4) | **9.3** (0.2) |
| musk | 123.9 (6.5) | 33.9 (1.3) | 101.3 (11.6) | 24.4 (1.9) | 143.4 (5.5) | **17.0** (0.7) |
| FICO | 13.3 (4.1) | 23.2 (1.4) | 8.7 (0.4) | **4.8** (0.3) | 88.1 (7.0) | 155.0 (27.5) |

The superiority of CG compared to BRS, AM, and BCD is carried over into Table 1, especially for larger datasets (bottom partition in the table). Compared to RIPPER, which is designed to maximize accuracy, CG is very competitive. The head-to-head "win-loss" record is nearly even and on no dataset is CG less accurate by more than 1%, whereas RIPPER is worse by $\sim 2\%$ on ionosphere, liver, and tic-tac-toe. Moreover on larger datasets, CG tends to learn significantly simpler rule sets that are nearly as or even more accurate than RIPPER, e.g. on bank-marketing, magic, and FICO. CART on the other hand is less competitive in this experiment. Tic-tac-toe is notable in admitting an exact rule set solution, corresponding to all positions with three x's or or's in a row. CG succeeds in finding this rule set whereas the other algorithms including RF cannot quite do so.

Given our use of IP, a relevant question is whether certifiably optimal or near optimal solutions to the Master IP can be obtained in practice. Such guarantees are most interesting when the achieved training accuracies are low as they rule out the existence of much better solutions. Among the small instances where the training accuracy is below 90% for CG, we are able to obtain optimal or near

optimal solutions to the training problem for heart, liver, and transfusion. For example, for transfusion, we can certify that the optimality gap is at most 0.7% when the bound on the complexity of the rule set $C$ is set to 15. Note that our IP formulation (1)-(4) solves the training problem with the Hamming loss objective. For the medium to large datasets, we are unable to accomplish this task as we are unable to solve the Pricing problem to optimality or near-optimality within our specified time limits.

We conclude this section with an example of a DNF rule learned by CG, specifically the one that maximizes accuracy on the FICO data with two simple clauses:

$$\left(\text{NumSatTrades} \geq 23\right) \wedge \left(\text{ExtRiskEstimate} \geq 70\right) \wedge \left(\text{NetFracRevolvBurden} \leq 63\right)$$

OR

$$\left(\text{NumSatTrades} \leq 22\right) \wedge \left(\text{ExtRiskEstimate} \geq 76\right) \wedge \left(\text{NetFracRevolvBurden} \leq 78\right).$$

According to the data dictionary provided with the FICO challenge [1], "NumSatTrades" is the number of satisfactory accounts, "ExtRiskEstimate" is a consolidated version of some risk markers, and "NetFracRevolvBurden" is the ratio of revolving balance to credit limit. The rules thus identify two groups, one with more accounts and less revolving debt, the other with fewer accounts and somewhat more revolving debt. A slightly higher (better) "ExtRiskEstimate" is required for the second, riskier group.

## 5 Conclusion

We have developed a column generation algorithm for learning interpretable DNF or CNF classification rules that efficiently searches the space of rules without pre-mining or other restrictions. Experiments have borne out the superiority of the accuracy-rule simplicity trade-offs achieved.

While the results in Table 1 are competitive with RIPPER, in some instances they fall short of the potential suggested in the first accuracy-complexity trade-off experiment. For example on the heart disease dataset, Figure 1a shows a maximum accuracy of 81.3% while the value resulting from CV in Table 1 is only 78.9%. For small datasets, the challenge is variability in estimating test accuracy. For large datasets, although we have proposed measures such as time limits and sampling to reduce the computational burden, these measures are applied more aggressively during cross-validation when many more instances need to be solved, thus affecting solution quality. We leave as future work improved procedures for optimizing parameter $C$ for accuracy.

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
