[Supplementary Material]

# A  Supplementary material

## A.1  Datasets and data processing

The UCI repository datasets were used largely as-is. We note the following deviations and label binarizations:

- Liver disorders: We used the number of drinks as the output variable as recommended by the data donors rather than the selector variable. The number of drinks was binarized as either $\leq 2$ or $> 2$.

- Gas sensor array drift: The label was binarized as either $\leq 3$ or $> 3$ as in [19].

- Heart disease: We used only the Cleveland data and removed 4 samples with 'ca' = ?, yielding 299 samples. The label was binarized as either 0 or $> 0$ as in other works.

For the FICO dataset, missing and special values were processed as follows. First, 588 records with all entries missing (values of $-9$) were removed. Values of $-7$ (no inquiries or delinquencies observed) were replaced by the maximum number of elapsed months in the data plus 1. Values of $-8$ (not applicable) and remaining values of $-9$ (missing) were combined into a single null category. During binarization, a special indicator was created for these null values and all other comparisons with the null value return False. Values greater than 7 (other) in 'MaxDelq2PublicRecLast12M' were imputed as 7 (current and never delinquent) based on the corresponding values in 'MaxDelqEver'.

## A.2  BRS parameters

We followed [52] and its associated code in setting the parameters of BRS and FPGrowth, the frequent rule miner that BRS relies on: minimum support of 5% and maximum length 3 for FPGrowth; reduction to 5000 candidate rules using information gain (this reduction was triggered in all cases); $\alpha_+ = \alpha_- = 500$, $\beta_+ = \beta_- = 1$, and 2 simulated annealing chains of 500 iterations for BRS itself.

## A.3  Accuracy-simplicity trade-offs for all datasets

Below in Figures 2 and 3 is the full set of accuracy-simplicity trade-off plots for all 16 datasets, including the 4 from the main text.

## A.4  Results for additional classifiers

As discussed in the main text, we were unable to execute code from the authors of Interpretable Decision Sets (IDS) [33] with practical running time when the number of candidate rules mined by Apriori [2] exceeded 1000. While it is possible to limit this number by increasing the minimum support and decreasing the maximum length parameters of Apriori, we did not do so beyond a support of 5% and length of 3 (same values as with FPGrowth for BRS) as it would severely constrain the resulting candidate rules. Thus we opted to run IDS only on those datasets for which Apriori generated fewer than 900 candidates given minimum support of 5% and either maximum length of 3 or unbounded length.

In terms of the settings for IDS itself, we ran a deterministic version of the local search algorithm with $\epsilon = 0.05$ as recommended by the authors. We set $\lambda_6 = \lambda_7 = 1$ to have equal costs for false positive and negatives, consistent with the other algorithms. For simplicity, the overlap parameters $\lambda_3$ and $\lambda_4$ were set equal to each other and tuned separately for accuracy, yielding $\lambda_3 = \lambda_4 = 0.5$. $\lambda_5$ was set to 0 as it is not necessary for binary classification. Lastly, $\lambda_1$ and $\lambda_2$ were set equal to each other to reflect the choice of complexity metric as the number of rules plus the sum of their lengths. We then varied $\lambda_1 = \lambda_2$ over a range to trade accuracy against complexity.

Our partial results for IDS are shown in Tables 3 and 5. Despite "cheating" in the sense of choosing $\lambda_1 = \lambda_2$ to maximize accuracy after all the test results were known, the performance is not competitive with the other rule set algorithms on most datasets. In addition to the constraints placed on Apriori, we suspect that another reason is that the IDS implementation available to us is designed primarily for the interval representation of numerical features (see Section 4) and is not easily adapted to handle the alternative $(\leq, >)$ representation.

(a) banknote

(b) heart

(c) ILPD

(d) ionosphere

(e) liver

(f) pima

(g) tic-tac-toe

(h) transfusion

Figure 2: Rule complexity-test accuracy trade-offs. Pareto efficient points are connected by line segments.

(a) WDBC

(b) adult

(c) bank-marketing

(d) gas

(e) magic

(f) mushroom

(g) musk

(h) FICO

Figure 3: Rule complexity-test accuracy trade-offs. Pareto efficient points are connected by line segments.

Table 3: Mean test accuracy for rule set classifiers (%, standard error in parentheses)

| dataset | CG | BRS | AM | BCD | IDS | RIPPER |
|---|---|---|---|---|---|---|
| banknote | 98.8 (1.2) | 99.1 (0.2) | 98.5 (0.4) | 98.7 (0.2) | 65.2 (2.1) | 99.2 (0.2) |
| heart | 78.9 (2.4) | 78.9 (2.4) | 72.9 (1.8) | 74.2 (1.9) | | 79.3 (2.2) |
| ILPD | 69.6 (1.2) | 69.8 (0.8) | 71.5 (0.1) | 71.5 (0.1) | 71.5 (0.1) | 69.8 (1.4) |
| ionosphere | 90.0 (1.8) | 86.9 (1.7) | 90.9 (1.7) | 91.5 (1.7) | | 88.0 (1.9) |
| liver | 59.7 (2.4) | 53.6 (2.1) | 55.7 (1.3) | 51.9 (1.9) | 51.0 (0.2) | 57.1 (2.8) |
| pima | 74.1 (1.9) | 74.3 (1.2) | 73.2 (1.7) | 73.4 (1.7) | 68.4 (0.9) | 73.4 (2.0) |
| tic-tac-toe | 100.0 (0.0) | 99.9 (0.1) | 84.3 (2.4) | 81.5 (1.8) | | 98.2 (0.4) |
| transfusion | 77.9 (1.4) | 76.6 (0.2) | 76.2 (0.1) | 76.2 (0.1) | 76.2 (0.1) | 78.9 (1.1) |
| WDBC | 94.0 (1.2) | 94.7 (0.6) | 95.8 (0.5) | 95.8 (0.5) | 85.1 (2.2) | 93.0 (0.9) |
| adult | 83.5 (0.3) | 81.7 (0.5) | 83.0 (0.2) | 82.4 (0.2) | | 83.6 (0.3) |
| bank-mkt | 90.0 (0.1) | 87.4 (0.2) | 90.0 (0.1) | 89.7 (0.1) | | 89.9 (0.1) |
| gas | 98.0 (0.1) | 92.2 (0.3) | 97.6 (0.2) | 97.0 (0.3) | | 99.0 (0.1) |
| magic | 85.3 (0.3) | 82.5 (0.4) | 80.7 (0.2) | 80.3 (0.3) | 72.0 (0.1) | 84.5 (0.3) |
| mushroom | 100.0 (0.0) | 99.7 (0.1) | 99.9 (0.0) | 99.9 (0.0) | | 100.0 (0.0) |
| musk | 95.6 (0.2) | 93.3 (0.2) | 96.9 (0.7) | 92.1 (0.2) | | 95.9 (0.2) |
| FICO | 71.7 (0.5) | 71.2 (0.3) | 71.2 (0.4) | 70.9 (0.4) | | 71.8 (0.2) |

Table 4: Mean test accuracy for other classifiers (%, standard error in parentheses)

| dataset | CART | RF | LR | SVM |
|---|---|---|---|---|
| banknote | 96.8 (0.4) | 99.5 (0.1) | 98.8 (0.2) | 99.9 (0.1) |
| heart | 81.6 (2.4) | 82.5 (0.7) | 83.6 (2.5) | 82.9 (1.9) |
| ILPD | 67.4 (1.6) | 69.8 (0.5) | 72.9 (0.8) | 70.8 (0.6) |
| ionosphere | 87.2 (1.8) | 93.6 (0.7) | 86.9 (2.6) | 94.9 (1.8) |
| liver | 55.9 (1.4) | 60.0 (0.8) | 59.1 (2.0) | 59.4 (1.7) |
| pima | 72.1 (1.3) | 76.1 (0.8) | 77.9 (1.9) | 76.8 (1.9) |
| tic-tac-toe | 90.1 (0.9) | 98.8 (0.1) | 98.3 (0.4) | 98.3 (0.4) |
| transfusion | 78.7 (1.1) | 77.3 (0.3) | 77.0 (0.8) | 77.0 (0.3) |
| WDBC | 93.3 (0.9) | 97.2 (0.2) | 95.4 (0.9) | 98.2 (0.4) |
| adult | 83.1 (0.3) | 84.7 (0.1) | 85.1 (0.2) | 84.8 (0.2) |
| bank-mkt | 89.1 (0.2) | 88.7 (0.0) | 89.8 (0.1) | 88.7 (0.0) |
| gas | 95.4 (0.1) | 99.7 (0.0) | 99.4 (0.1) | 99.5 (0.1) |
| magic | 82.8 (0.2) | 86.6 (0.1) | 79.0 (0.2) | 87.7 (0.3) |
| mushroom | 96.2 (0.3) | 99.9 (0.0) | 99.9 (0.1) | 100.0 (0.0) |
| musk | 90.1 (0.3) | 86.2 (0.4) | 93.1 (0.2) | 97.8 (0.1) |
| FICO | 70.9 (0.3) | 73.1 (0.1) | 71.6 (0.3) | 72.3 (0.4) |

In Table 4, accuracy results of logistic regression (LR) and support vector machine (SVM) classifiers are included along with those of non-rule set classifiers from the main text (CART and RF). Although LR is a generalized linear model, it may not be regarded as interpretable in many application domains. For SVM, we used a radial basis function (RBF) kernel and tuned both the kernel width as well as the complexity parameter $C$ using nested cross-validation.

Table 5: Mean complexity (# clauses + total # conditions, standard error in parentheses)

| dataset | CG | BRS | AM | BCD | IDS | RIPPER | CART |
|---|---|---|---|---|---|---|---|
| banknote | 27.8 (1.6) | 30.4 (1.1) | 24.2 (1.5) | 21.3 (1.9) | 11.2 (0.5) | 28.6 (1.1) | 51.8 (1.4) |
| heart | 11.3 (1.8) | 24.0 (1.6) | 11.5 (3.0) | 15.4 (2.9) | | 16.0 (1.5) | 32.0 (8.1) |
| ILPD | 10.9 (2.7) | 4.4 (0.4) | 0.0 (0.0) | 0.0 (0.0) | 2.0 (0.0) | 9.5 (2.5) | 56.5 (10.9) |
| ionosphere | 12.3 (3.0) | 12.0 (1.6) | 16.0 (1.5) | 14.6 (1.4) | | 14.6 (1.2) | 46.1 (4.2) |
| liver | 5.2 (1.2) | 15.1 (1.3) | 8.7 (1.8) | 4.0 (1.1) | 0.0 (0.0) | 5.4 (1.3) | 60.2 (15.6) |
| pima | 4.5 (1.3) | 17.4 (0.8) | 2.7 (0.6) | 2.1 (0.1) | 6.0 (0.3) | 17.0 (2.9) | 34.7 (5.8) |
| tic-tac-toe | 32.0 (0.0) | 32.0 (0.0) | 24.9 (3.1) | 12.6 (1.1) | | 32.9 (0.7) | 67.2 (5.0) |
| transfusion | 5.6 (1.2) | 6.0 (0.7) | 0.0 (0.0) | 0.0 (0.0) | 2.0 (0.0) | 6.8 (0.6) | 14.3 (2.3) |
| WDBC | 13.9 (2.4) | 16.0 (0.7) | 11.6 (2.2) | 17.3 (2.5) | 15.2 (0.7) | 16.8 (1.5) | 15.6 (2.2) |
| adult | 88.0 (11.4) | 39.1 (1.3) | 15.0 (0.0) | 13.2 (0.2) | | 133.3 (6.3) | 95.9 (4.3) |
| bank-mkt | 9.9 (0.1) | 13.2 (0.6) | 6.8 (0.7) | 2.1 (0.1) | | 56.4 (12.8) | 3.0 (0.0) |
| gas | 123.9 (6.5) | 22.4 (2.0) | 62.4 (1.9) | 27.8 (2.5) | | 145.3 (4.2) | 104.7 (1.0) |
| magic | 93.0 (10.7) | 97.2 (5.3) | 11.5 (0.2) | 9.0 (0.0) | 10.0 (0.0) | 177.3 (8.9) | 125.5 (3.2) |
| mushroom | 17.8 (0.3) | 17.5 (0.4) | 15.4 (0.6) | 14.6 (0.6) | | 17.0 (0.4) | 9.3 (0.2) |
| musk | 123.9 (6.5) | 33.9 (1.3) | 101.3 (11.6) | 24.4 (1.9) | | 143.4 (5.5) | 17.0 (0.7) |
| FICO | 13.3 (4.1) | 23.2 (1.4) | 8.7 (0.4) | 4.8 (0.3) | | 88.1 (7.0) | 155.0 (27.5) |