[Reviews · NeurIPS 2018]

Reviewer 1



The authors propose a mathematical programming approach to build interpretable machine learning models. In this case, the interpretable model is a system of Boolean rules in disjunctive (or conjunctive) normal form which is constructed using column generation for the linear relaxation of a mixed integer program (MIP) designed to minimize the number of positive samples classified incorrectly and the complexity of the learned system subject to complexity constraints. To remedy the fact that there are exponentially many potential clauses to optimize over, the authors propose a standard column generation approach that prices potential columns to add and solves a secondary integer program to find such potential columns. The authors also note that the column generation can also be done via heuristics or a greedy algorithm. Once the linear programming program is solved or reaches its time limit, the approach then solves the global mixed integer formulation to get a final set of rules. The authors perform many experiments comparing their model to a wide range of benchmark models testing improvements in model complexity as well as accuracy. Several benchmark models are built with only one of these two objectives in mind and the proposed model seems to give good tradeoffs in both objectives. The goal of having an interpretable decision making model is very well motivated. Additionally, the proposed methodology is very clearly explained and the experimental design is nicely laid out. Overall, the column generation mechanism seems to scale better than previous work and the mathematical programming formulation allows easy access to bound the optimal accuracy and complexity. Furthermore, the mathematical programming formulation is very flexible and allows for various loss formulations as well as differing tradeoffs between accuracy and complexity. This methodology is seemingly extensible enough to account for other desired traits of a machine learning model such as fairness constraints. The approach can also be extended to explicitly trade off complexity and accuracy. Finally, the approach seems to do very well against several benchmark in terms of test accuracy and model complexity. For these reasons, I recommend that this paper be accepted, and hope the authors can improve it by addressing the following questions/suggestions: - The MIP is missing bounds on the 0 <= \xi_i <= 1 (same for \delta_i in the pricing problem). While you are right that you don’t need \xi_i \in \{0,1\}, if you don’t include lower/upper bounds, then if for a given i in constraint (2), the sum of w_k variables is large, then the optimal solution would make the corresponding \xi_i negative (to make the objective smaller) while maintaining feasibility of constraint (2). I think you probably have the bounds right in the implementation, but not in the text. - Derivation of the pricing problem: since this IP approach may be new to many NIPS readers, it may be better to give a more gentle derivation of the pricing problem, defining reduced costs; just a suggestion. - Tables 1 and 2 are somewhat difficult to read and would benefit from either bolding the best value in the row or writing the value of the best entry in a final column. It would also be helpful to explain some of the more interesting rows such as the tic-tac-toe dataset having 100 test accuracy with 0.0 standard error for CG while the random forest has less accuracy. This could be resolved by reporting training accuracy or demonstrating that the CG approach enables better generalization due to its lack of complexity. - It would be helpful to demonstrate the ability for the system to give lower bounds on complexity by reporting these metrics for the different experiments. - Can you elaborate on the FICO example end of page 7? What are these features, and why is this an intuitive rule? Without a few sentences explaining it, the example is kind of useless. - The related work section could include brief references to work that uses column generation for machine learning tasks, examples below. [1] Demiriz, Ayhan, Kristin P. Bennett, and John Shawe-Taylor. "Linear programming boosting via column generation." Machine Learning 46.1-3 (2002): 225-254. [2] Bi, Jinbo, Tong Zhang, and Kristin P. Bennett. "Column-generation boosting methods for mixture of kernels." Proceedings of the tenth ACM SIGKDD international conference on Knowledge discovery and data mining. ACM, 2004. [3] Li, Xi, et al. "Learning hash functions using column generation." arXiv preprint arXiv:1303.0339 (2013).

Reviewer 2



The paper provides an appealing approach to learning boolean rules: encode the problem as an IP and use column generation to cope with the large number of variables needed (in the given formulation). A nice aspect of this approach is that the desired trade-off between accuracy and complexity is explicitly represented. (Mostly) positive empirical results are presented. The key issue, of course, is the complexity of the pricing problem. Here we do not get a theoretical result on this, but I suspect it is NP-hard. Instead we get empirical observations that (1) the pricing problem can get quite big and (2) thus not always solvable to optimality. Indeed for big problems the pricer can fail to return any negative reduced cost variables to add to the master problem. The authors deal with these problems in a reasonable way: resorting to approximate methods. Since the authors are using CPLEX they can encode constraints (7) and (8) using CPLEX's "Logical AND" constraint. I strongly expect this to give better performance than "manually" encoding it using linear constraints as is presented here. This is because (I assume - and hope!) that CPLEX will have specialised propagation for logical AND constraints. One could argue that this paper is "too easy": the authors encode the problem as an IP with CG and get CPLEX to do most of the hard work. But, really, this very easiness is the point. I think the authors are clear that further work is needed to get the most out of this technique which is true and good to see explicitly stated. The paper is well-written and I found no typos (which is unusual). One small point: an overall time limit of 300 seconds seems low. If I were, say, a bank wanting a comprehensible rule set for fraud detection I would happily wait a few days to get it. AFTER DISCUSSION AND FEEDBACK I have bumped up my rating a notch. Nice work.

Reviewer 3



The authors propose an integer linear program formulation to learn boolean formulas for classification tasks in a supervised setting. To solve the ILP, a Dantzig Wolfe decomposition is used for its LP-relaxation. Experimental comparison is perfomed on some small scale datasets where explainable classifiers could be advantageous. Strengths: - The paper is well-written. - The problem formulation is, to my knowledge, novel and has theoretical appeal due to its simplicity and combinatorial structure. - The proposed algorithm for solving the intractable optimization problem is sound. - Experimental comparison is performed against a large set of competitors and satisfactory results are obtained. - Weaknesses: - The authors approach is only applicable for problems that are small or medium scale. Truly large problems will overwhelm current LP-solvers. - The authors only applied their method on peculiar types of machine learning applications that were already used for testing boolean classifier generation. It is unclear whether the method could lead to progress in the direction of cleaner machine learning methods for standard machine learning tasks (e.g. MNIST). Questions: - How where the time limits in the inner and outer problem chosen? Did larger timeouts lead to better solutions? - It would be helpful to have an algorithmic writeup of the solution of the pricing problem. - SVM gave often good results on the datasets. Did you use a standard SVM that produced a linear classifier or a Kernel method? If the former is true, this would mean that the machine learning tasks where rather easy and it would be necessary to see results on more complicated problems where no good linear separator exists. Conclusion: I very much like the paper and strongly recommend its publication. The authors propose a theoretically well grounded approach to supervised classifier learning. While the number of problems that one can attack with the method is not so large, the theoretical (problem formulation) and practical (Dantzig-Wolfe solver) contribution can possibly serve as a starting point for further progress in this area of machine learning.